# State of the Art Technologies for High Yield Heterologous Expression and Production of Oxidoreductase Enzymes: Glucose Oxidase, Cellobiose Dehydrogenase, Horseradish Peroxidase, and Laccases in Yeasts *P. pastoris* and *S. cerevisiae*

Milica Crnoglavac Popović [ID], Marija Stanišić and Radivoje Prodanović *[ID]

Faculty of Chemistry, University of Belgrade, Studentski trg 12-16, 11000 Belgrade, Serbia; milicac@chem.bg.ac.rs (M.C.P.); mstanisic@chem.bg.ac.rs (M.S.)
* Correspondence: rprodano@chem.bg.ac.rs; Tel.: +381-638708018

**Abstract:** Oxidoreductase (OXR) enzymes are in high demand for biocatalytic applications in the food industry and cosmetics (glucose oxidase (GOx) and cellobiose dehydrogenase (CDH)), bioremediations (horseradish peroxidase (HRP) and laccase (LAC)), and medicine for biosensors and miniature biofuel cells (GOx, CDH, LAC, and HRP). They can be used in a soluble form and/or within the yeast cell walls expressed as chimeras on the surface of yeast cells (YSD), such as *P. pastoris* and *S. cerevisiae*. However, most of the current studies suffer from either low yield for soluble enzyme expression or low enzyme activity when expressed as chimeric proteins using YSD. This is always the case in studies dealing with the heterologous expression of oxidoreductase enzymes, since there is a requirement not only for multiple OXR gene integrations into the yeast genome (super transformations), and codon optimization, but also very careful design of fermentation media composition and fermentation conditions during expression due to the need for transition metals (copper and iron) and metabolic precursors of FAD and heme. Therefore, scientists are still trying to find the optimal formula using the above-mentioned approaches; most recently, researcher started using protein engineering and directed evolution to increase in the yield of recombinant enzyme production. In this review article, we will cover all the current state-of-the-art technologies and most recent advances in the field that yielded a high expression level for some of these enzymes in specially designed expression/fermentation systems. We will also tackle and discuss new possibilities for further increases in fermentation yield using cutting-edge technologies such as directed evolution, protein and strain engineering, high-throughput screening methods based on in vitro compartmentalization, flow cytometry, and microfluidics.

**Keywords:** recombinant; oxidoreductase; expression; yeasts; directed evolution; high-throughput screening; flow cytometry; in vitro compartmentalization

## 1. Introduction

Enzymes are biocatalysts with complex structures and specific catalytic mechanisms that determine their distinctive properties, such as high catalytic activity and selectivity of specific substrates. According to the BRENDA database, oxidation–reduction reactions constitute at least thirty percent of all enzymatic reactions; given that fact, oxidoreductase (OXR) occupies a special place among biocatalysts [1–3]. These enzymes catalyze the transfer of electrons from an electron donor to an electron acceptor molecule. Different cofactors, such as heme, flavin, and metal ions, are necessary for OXR catalytic activity [4]. That usually complicates their expression. The abundance of these enzymes is versatile, and the source of OXR defines their biological functions. These enzymes act as efficient biocatalysts in various processes and fields of biotechnology and have a wide range of applications in the degradation of xenobiotic compounds, the design of

biosensors for environmental or medical purposes, the food and textile industry, and other fields [3].

## 1.1. Glucose Oxidase (GOx)

Glucose oxidase (EC 1.1.3.4) is an enzyme belonging to the OXR group. This flavoprotein uses molecular oxygen as an electron acceptor in a two-step reaction to catalyze the oxidation of β-D-glucose to D-glucono-δ-lactone and $H_2O_2$. In the first half of the reaction, namely reduction, GOx catalyzes the oxidation of β-D-glucose to D-glucono-δ-lactone, which is then hydrolyzed to gluconic acid. During the first phase cofactor of GOx, flavine adenine dinucleotide (FAD) is reduced to $FADH_2$. In the second phase, the oxidative half-reaction, oxygen reoxidizes the reduced GOx to produce $H_2O_2$ and $FADH_2$ oxidizes to FAD [5]. In 1999, the structure of glucose oxidase from *Aspergillus* was confirmed, which determined that GOx consists of two uniform subunits and that both subunits contain two separate domains: one is not covalently bound with FAD and the second attaches to the substrate. Structurally, the first domain is mainly a β-sheet and the second domain consists of four α-helices and an antiparallel β-sheet [6,7]. GOx is an enzyme that is considered an "ideal enzyme". It is often called an oxidase "Ferrari" because it has high activity, stability, and specificity and can be used in various biotechnological, medical, and industrial applications [8,9].

## 1.2. Cellobiose Dehydrogenase (CDH)

Cellobiose dehydrogenase (CDH; EC1.1.99.18; CAZy AA3_1) belongs to the OXR enzyme family [10]. Many species of wood-decaying fungi produce this glycosylated enzyme involved in lignocellulose degradation [11]. CDH is usually monomeric and consists of two domains, a C-terminal cytochrome-binding fragment (CYT) and a catalytic flavin-containing dehydrogenase domain (DH), which are connected via a linear papain-sensitive linker peptide [12]. The catalytic mechanism of CDH is comprised of an oxidative and reductive half-reaction. The substrate is converted into appropriate lactone during the oxidative half-reaction while FAD is reduced to $FADH_2$. Subsequently, electrons are transferred to the electron acceptor [13]. Intensive investigation has been carried out in the last decade on this enzyme because of its possible application in many fields like biosensors, biofuel cells, bioremediation, and clinical application [10,14].

## 1.3. Horseradish Peroxidase (HRP)

Horseradish peroxidase (EC 1.11.1.7) belongs to the group of OXR. This glycoprotein, with 44 kDa, contains at least 15 distinct isoenzyme forms, the most common and, thus, most studied of which is isoenzyme C1A [15]. It was determined that the tertiary structure of HRP includes two domains formed by ten α-helices, four disulfide bonds, two $Ca^{2+}$ ions per molecule, and the prosthetic heme group [16–18]. The reaction between $H_2O_2$ and Fe(III) in the active center is the first phase of the catalytic cycle. As a result of this phase, a high oxidation state intermediate consisting of a Fe(IV) oxoferryl center and porphyrin-based cation radical, called Compound I, is generated. Compound II, which represents Fe(IV) oxoferryl species, is generated in the first one-electron reduction step and reduced in the second step by producing an enzyme resting state. The excess hydrogen peroxide reacts with the resting state enzyme and, as a product, compound III is obtained [19]. Due to the increasing possibilities for the applications of this enzyme in biotechnology and other branches of industry, solving the problem of recombinant production of this enzyme represents a significant challenge in science.

## 1.4. Laccase (LAC)

Laccases (benzenediol: oxygen oxidoreductase; p-diphenol oxidase EC 1.10.3.2) belong to a large family of multicopper oxidases. Like all multicopper oxidases, LAC also possess a relatively uncomplicated 3D structure, mainly comprising beta sheets and thurns, including a small cupredoxin-like domain. They are glycoproteins with four Cu

atoms per monomer. These four Cu atoms form the catalytic core, which helps the enzyme catalyze the redox reaction. LAC couples the four single electron oxidations of a reducing substrate to the four-electron reduction cleavage of the dioxygen bond using four Cu atoms [20]. LACs are some of the oldest and most widespread enzymes in nature, and they can be found in fungi, bacteria, plants, and animals, and their function depends on the biological source. The reactions performed by LAC include the rupture of alkyl-aryl bonds, the oxidation of benzyl alcohol, and the rupture of aromatic rings that generate a wide variety of oxidized phenolic compounds. In addition, in vitro studies have shown that LACs are capable of polymerization, depolymerization, methylation, and demethylation reactions, as well as oxidation of o- and p-diphenols, aminophenols, polyphenols, polyamines, aryl-amines, and several other phenolic compounds [21–24]. The scope of laccase-catalyzed reactions can be expanded using mediators. The wide substrate spectrum and molecular oxygen as a final electron acceptor with the water molecule as the only by-product make LAC "eco-friendly" and attractive for various biotechnological industries. These biocatalysts have several bioremediation and biodegradation applications in numerous industries (food, cosmetics, nanobiotechnological, textile, woodworking, and pulp/paper) [20,25,26].

The production of oxidoreductases from native sources cannot meet the high market demand due to low yields and the incompatibility of the standard industrial fermentation processes with the conditions required for the growth of many microorganisms [6]. Recombinant technologies can be used to achieve higher yields of these enzymes. The diversity and upscaling possibilities of heterologous protein expression opened new commercial opportunities for their industrial uses [6].

The main goal of this review article is a detailed overview of the most recent trials of the heterologous expression of GOx, CDH, HRP, and LAC, as well as the main principles and problems drawn from these trials. A discussion of the expression systems based on yeast to OXR will be given since they offer a compromise between simplicity and the high protein expression yield offered by procaryotic expression systems (*E. coli* and *B. subtilis*) and post-translational modifications offered by mammalian expression systems (too complex and low protein yield) often needed for eukaryotic proteins.

In the end, some of the state-of-the-art technologies that could be used to increase the expression of these OXRs, like directed evolution, strain engineering, protein engineering, high-throughput screening, flow cytometry, microfluidics, and in vitro compartmentalization, will be given, as well as possible future research directions.

## 2. Heterologous Expression

Various expression systems exist for recombinant protein production in bacteria, yeasts, insects, and mammalian cells. All of them have their drawbacks and advantages. Yeasts are interesting and versatile hosts caused by benefits such as growth speed, simple genetic manipulations, secretory expression, post-translational modification, scalable fermentation, high biomass concentrations, and safe, pathogen-free production [27,28]. There are two large yeast expression systems: methylotroph and non-methylotroph. Typical examples of non-methylotroph and methylotroph yeasts are *Saccharomyces cerevisiae* and *Pichia pastoris*.

### 2.1. Saccharomyces Cerevisiae

*Saccharomyces cerevisiae*, known as baker's yeast, was the first eukaryotic organism with a completely sequenced genome [29]. As said before, *S. cerevisiae* belongs to the non-methylotroph yeast group. It was initially developed as a replacement host for producing a recombinant protein that could not be expressed in bacterial cells [27,30]. The native resistance of *S. cerevisae* to low pH, high osmolality, and numerous inhibitors (acetic acid, furfural, vanillin, etc.) allows low-cost and facile fermentation procedures with high biomass concentrations under aerobic and anaerobic conditions, thereby enabling a higher yield of recombinant proteins [31]. Plentiful genetic tools are developed for expression

in *S. cerevisiae*, such as recombinant protein expression controlled by strong constitutive promoters like *TEF1* and *GAP* or inducible promoters like galactose-inducible *GAL1* [32]. Unfortunately, this expression host has some disadvantages such as a low level of protein secretion, hyperglycosylation [33], and proteolytic degradation of expressed proteins. From a biotechnological point of view, this expression system is "generally recognized as safe" (GRAS) because it is nonpathogenic and has a history in the food and pharmaceutical industry [27].

With the intention of achieving a higher expression level, a few approaches were used, such as various fermentation conditions, optimization of the codon, strong promotors and terminators, and a multi-copy expression vector—Table 1 [27,34,35].

**Table 1.** Overview of the fermentation experiments and parameters used to optimize recombinant GOx, CDH, LAC, and HRP expression in *S. cerevisiae*, including the fermentation yield of the enzyme.

| Source | Variant of Oxidoreductase Enzyme | Host Strain; Vector | Promoter | Inducer | Signal Sequence | Additional Information | Enzyme Yield | Ref. |
|---|---|---|---|---|---|---|---|---|
| **GOx** | | | | | | | | |
| *A. niger* | NR | 2805; YEp352 | *GAL1* | 1% galactose | ss of α-factor | NR | 32 [a] U/mL | [36] |
| *A. oryzae* | NR | 2805 | *GAL-10* | NR | α-amylase signal sequence | 30 °C, 150 rpm, feedback-controlled fed-batch | NR | [37] |
| *A. niger* | NR | 2805; Yep352 | *Hybrid ADH2-GPD* | 2% glucose | ss of α-factor | 1.5% EtOH | 260 [a] U/mL | [36] |
| *A. niger* | NR | GRF181 pSGO2 | *ADH2-GPD* | 8% sucrose | Native | Shake flask; 28 °C; 200 h | 106 [a] U/mL | [38] |
| **CDH** | | | | | | | | |
| *M. thermophilum* | Wild type | BJ5465; pJRoC30 | *GAL1* | 2% galactose | Native | Deep-well plate (500 µL of medium); 30 °C; 5 days | 50 [b] U/L | [39] |
| *M. thermophilum* | Wild type | BJ5465; pJRoC30 | *GAL1* | 2% galactose | ss of α factor | Deep-well plate (500 µL of medium); 30 °C; 5 days | 16 [b] U/L | [39] |
| *P. chrysosporium* | U46081.1 | InvSC1; pYES2 | *GAL1* | Galactose | Native | Shake flask; 30 °C; 16 h | NR | [40] |
| *T. clypeatus* | GAFV01008428.1 | BY4742; pFL61 | *PGK* | No | No | NR type of cultivation; Czapek medium, 3 days | 0.039 [b] U/mg | [41] |
| **HRP** | | | | | | | | |
| *Horseradish* | Wild type | SIP-Ost1 (Δ44–70); modified pESC-URA | *TDH3* | | pre-Ost1 | Fermenter 5 L (batch fermentation) | 13,506 [c] U/L | [42] |
| *Horseradish* | HRP 3-17E12 | BJ5465; pYEX-S1 | *PGK1* | No | NR | Expression time 25 h | about 250 [c] U/L | [43] |
| **LAC** | | | | | | | | |
| *M. thermophila* | MtL | BJ5465; pJRoC3 | NR | NR | NR | Shake flask 2.8 L; 0.005 mM CuSO₄ 30 °C; 1 day | 0.6 [d] U/L | [44] |
| *M. thermophila* | T2 mutant | BJ5465; pJRoC3 | NR | NR | NR | Shake flask 2.8 L; 0.005 mM CuSO₄; 30 °C; 1 day | 102 [d] U/L | [44] |
| *T. versicolor* | Cvl3 | BY2777; pYES2 | *GAL1* | 4% Galactose | Native | Shake flask 0.3 L; 0.5 mM CuSO₄; 20 °C; 6 days | 45 [e] U/L | [45] |
| *L. edodes* | Lcc4 | FGY217; pBG13 | *GAL1* | 4% Galactose | Native | Fermentor 4 L; 0.5 mM CuSO₄; 20 °C; 7 days | 10 [e] U/L | [46] |

**Table 1.** *Cont.*

| Source | Variant of Oxidoreductase Enzyme | Host Strain; Vector | Promoter | Inducer | Signal Sequence | Additional Information | Enzyme Yield | Ref. |
|---|---|---|---|---|---|---|---|---|
| *A. pediades* | ApL | BJ5465; pJRoC30 | *GAL1* | 2.2% Galactose | $\alpha_{9H2}$ signal peptide | Shake flask 0.1 L; 0.4 mM $CuSO_4$; 20 °C; 4 days | 280 [e] U/L | [47] |
| *Trametes* sp. *C30* | Clac1, 2, 3 | W303-1A; YIp351 | *PGK1* | No | ss of SUC2 gene product | Fermentor 3 L; 1 mM $CuSO_4$; 28 °C; 3 days | 1200 [d] U/L | [48] |
| *M. thermophila* | T2 mutant | BW31a; pVT-100U | *ADH1* | No | Native | Shake flask 0.25 L; 0.6 mM $CuSO_4$; 30 °C; 1 day; 0.8% alanine | 6.52 [e] U/L | [49] |
| *T. versicolor* | Lcc1 | BW31a; pVT-100U | *ADH1* | No | Native | Shake flask 0.25 L; 0.6 mM $CuSO_4$; 30 °C; 1 day; 0.8% alanine | 0.45 [e] U/L | [49] |
| *T. trogii* | Lcc1 | BW31a; pVT-100U | *ADH1* | No | Native | Shake flask 0.25 L; 0.6 mM $CuSO_4$; 20 °C; 14 days; 0.8% alanine | 14.12 [e] U/L | [49] |

NR, not reported; Ref., reference; ss, secretion signal. [a] GOx activity was determined using o-dianisidine. [b] CDH activity was determined using 2,6-dichlorophenolindophenol (DCIP). [c] HRP activity was determined using 2,2′-azino-bis(3-ethylbenzothiazoline-6-sulphonic) acid (ABTS). [d] LAC activity was determined using syringaldazine (SGZ). [e] LAC activity was determined using 2,2′-azino-bis(3-ethylbenzothiazoline-6-sulphonic) acid (ABTS).

### 2.1.1. GOx

In 1992, Baetselier et al. confirmed that expressing up to 1.5 g/L using a suitable vector–host system was possible [38]. In this study, they used expression vectors pαGO1 and pSGO2, which contain an expression cassette of the regulated hybrid promoter *ADH2-GAPDH*, the α-factor or GOx signal sequence for secretion, the mature GOx cDNA, and the *GAPDH* terminator. The yeast strain GRF181 transformants of these plasmids were cultured on a YP medium with 8% sucrose [38]. The bidirectional inducible promoters can control recombinant expression in *S. cerevisiae*, galactose dehydrogenase 1 (*GAL1*), and UDP-glucose-epimerase (*GAL10*). For the expression induced by the *GAL1* promoter, 1% galactose concentration was the most successful and was obtained at 32 U/mL [36]. It was also noted that indirect control of ethanol production in feedback controlled fed-batch processes could be accomplished via feeding carbon sources. Galactose feeding was utilized to control the pH of the culture, decrease ethanol production, and improve the quantity of GOx produced [37].

### 2.1.2. CDH

*Termitomyces clypeatus* CDH intracellular expression in *S. cerevisiae* was achieved by Banerjee et al. [41] in a relatively low but detectable degree of the expressed protein, 0.039 U/mg, with cellobiose as an electron donor [41]. Previous research showed that the secretion signal influences the level of cellobiose dehydrogenase expression. Hence, the native secretory leader sequence gave three times higher expression than a signal sequence of α-factor [39]. Also, in research conducted earlier in our laboratory, the ability of *S. cerevisiae* to produce cellobiose dehydrogenase was confirmed, but the yields of the expressed enzymes were not high [40]. The results of work [39,40], both using the native secretion signal of CDH, are very different due to the use of different yeast strains. In work [39], the protease-free *S. cerevisiae* BJ5465 strain was used that can give high amounts of protein after long fermentation times. In contrast, in work [40], we used the *S. cerevisiae* InvSc1 strain (Invitrogen, Waltham, MA, USA), which usually gives a maximum of 1 mg/mL of recombinant protein after 16 h of fermentation and is mostly

used for directed evolution experiments. Most prior research has shown that *S. cerevisiae* is not a suitable host for high-yield cellobiose dehydrogenase production. Still, it is a great host choice for protein engineering techniques due to its higher transformation efficiency.

### 2.1.3. HRP

Zhao et al. obtained 5093 U/L of HRP via strain engineering, such as selecting an appropriate cis-regulatory module (CRM) and substituting a core promoter sequence (CPS) and terminator. The yield is significantly increased up to 330 times by replacing the α-factor signal peptide with another signal peptide for transport of HRP in the endoplasmic reticulum [42]. In this work, the authors concluded that improper combinations of pro-peptide and pre-peptide sequences can lead to the accumulation of recombinant HRP within the endoplasmic reticulum. Therefore, the author's eight-propeptide variants of the α-factor signal peptide were tested to facilitate the export of HRP from the endoplasmic reticulum. To confirm the increased amount of expressed HRP, the enzyme was purified and deglycosylated using EndoH. Using this additional prepropeptide sequence engineering in a 5 L flask reactor, the authors obtained 13,506 U/L of the HRP.

### 2.1.4. LAC

Functional LAC expression in *S. cerevisiae* was described in the early nineties. Afterward, articles about the optimization of heterologous LAC expression were published [50]. Cassland et al. recognized several important factors affecting LAC expression: the cultivation temperature, the gene selected, and the selected *S. cerevisiae* strain [48,51]. Previous research showed that heterologous production of LAC can be upgraded using protein engineering methods, so mutated *M. thermophila* LAC showed 170-fold higher activity than the wild type [44]. In the 2013, a group of authors demonstrated the possibility of secretory production of initially intracellular LAC and considered aeration an important factor affecting LAC heterologous production [46]. Iimura et al. confirmed that aeration is an important factor besides inducer and copper concentration, so the LAC yield was nearly two times higher when the culture device was used. Aeration was accelerated (45 U/L in baffled flask and 80 U/L when the culture device was used) [45]. A recent study examined a few factors affecting LAC expression in *S. cerevisiae*. It concluded that expression depends on the gene source, used construct, temperature, pH value of cultivation media, and copper concentration. This research has provided evidence for adding alanine to maintain the pH value of cultivation media because it greatly impacts laccase expression. Also, they proved that lowering the temperature during expression from 30 to 20 °C yielded more than double the laccase activity in the identical cultivation period [49]. Several studies suggest that the secretion signal could greatly impact LAC expression in *S. cerevisiae*, so Aza et al. successfully used an improved $\alpha_{9H2}$ signal sequence in their work [47,52]. As was the case in other enzymes expressed in yeasts, the structure of the carbohydrate component of the enzyme synthesized by yeast differs from the structure of carbohydrate component of the native enzyme. Usually, hyperglycosylation in yeast, especially in *S. cerevisiae*, occurs (up to 50% per mass) that can increase stability but also decrease the specific activity of the recombinant enzyme.

### 2.2. Pichia Pastoris

*Pichia pastoris* (recently renamed as *Komagataella phaffii*) has gained an important role in recombinant protein production in the past several decades. *P. pastoris* can use methanol as a sole carbon source, given the fact it belongs to methylotroph yeasts. Undemanding genetic manipulation, high cell culture density, the capability of secreting recombinant proteins into the cell culture medium, unexacting purification of secreted proteins, eukaryotic post-translational modification, and stability of genetic constructs make this host convenient for the recombinant expression of proteins [53,54]. Promoters that control the heterologous expression of proteins could be constitutive (glyceraldehyde-3-phosphate dehydrogenase and *PGAP*) or inducible (alcohol oxidase, *AOX1*, and *AOX2*) [55]. Con-

sidering the existence of *PAOX1* and *PAOX2* in this host genome, there are three possible phenotypes of *P. pastoris*, specifically Mut + (wild-type methanol utilization; both of the alcohol oxidase enzymes, Aox1p and Aox2p, are functional), Muts (slow methanol utilization; disrupted Aox1p and functional Aox2p), and Mut- (no methanol utilization; both Aox1p and Aox2p are disrupted) [29]. According to Walsh and Walsh (2022), *P. pastoris* takes a superior position regarding *S. cerevisiae* in the field of recombinant protein production; the reason for that probably lies in the tightly regulated expression of both the intracellular and extracellular recombinant protein, achieving high cell density caused by the aerobic process of respiration and glycosylation like in eukaryotic cells [28,56]. *P. pastoris* is "generally recognized as safe" (GRAS) by the Food and Drug Administration (FDA) and suitable for diverse biotechnological applications.

Several strategies can be applied to achieve a high yield of recombinant proteins. The strategies cover codon optimization, choice of suitable host strains and expression vectors, gene copy number, insertion of the gene of interest under the control of the strong promoter and appropriate signal sequence, and optimization of fermentation conditions (temperature, incubation period, agitation, carbon source, and concentration of inducer)—Table 2 [28,57,58]. The stability of integrated genes within yeast genomes without selection pressures is high, but this can still lead to 1% transformant loss after each cell division cycle.

**Table 2.** Overview of the fermentation experiments and parameters used for the optimization of recombinant Gox, CDH, LAC, and HRP expression in *P. pastoris*, including the fermentation yield of the enzyme.

| Source | Variant of Oxidoreductase Enzyme | Host Strain; Vector | Promoter | Inducer | Signal Sequence | Additional Information | Enzyme Yield | Ref. |
|---|---|---|---|---|---|---|---|---|
| **GOx** | | | | | | | | |
| *A. niger* | GOx accc30161 | SMD1168;pGAPZαA | *GAP* | NR | ss of α-factor | 30 °C; pH 6 | 107.18 [a] U/mL | [59] |
| *A. niger* | GOxM | SMD1168; pPIC3.5 | *AOX1* | 1% MeOH | | 30 °C; 3 days, 220 rpm | 26.93 [a] U/mL | [60] |
| *A. niger* ATCC 9029 | - | GS115; pPIC9 | *AOX1* | 1% MeOH | | 28 °C; 225 rpm | NR | [61] |
| *A. niger* | M12 mutant | KM71H; pPICZαA | *AOX* | 0.5% MeOH | Proalpha sequence | Nine days of fermentation | 17.5 [b] U/mL | [62] |
| **CDH** | | | | | | | | |
| *M. thermophilum* | N700S mutant | X33; pPICZαA | *AOX1* | 0.5% MeOH | ss of α-factor & propeptide | Fermentor 7 L; 30 °C; 5 days | 1800 [c] U/L | [39] |
| *P. cinnabarinus* | Wild type | X33; pPICZαA | *AOX1* | 3% MeOH | ss of α-factor | Fermentor 1 L; 4 days | 7800 [c] U/L | [63] |
| *N. crassa strain FGSC 2489* | NC-cdh1 | X33; pPICZαB | *AOX1* | 1% MeOH | ss of α-factor | Shake flask 0.25 L; 30 °C; 1 day | 7451 [c] U/L | [64] |
| *P. chrysosporium* | Mutant | KM71H; pPICZαA | *AOX1* | 0.5% MeOH | ss of α-factor | Shake flask; 28 °C; 6 days | 950 [c] U/L | [64] |
| **HRP** | | | | | | | | |
| *Horseradish* | wild type | X-33; pPICZαB | *AOX1* | 0.5% MeOH | ss of α-factor | 30 °C; BMGY medium supplemented with 1% casamino acids; BMMY medium supplemented with 1.0 mM vitamin B1, 1.0 mM δ-ALA, and trace element mix; the highest yield in 80–90 h post-induction | 377 [d] U/mg | [43] |
| *Horseradish* | mutant HRP 2-13A10 | X-33; pPICZαB | *AOX1* | 0.5% MeOH | ss of α-factor | Same as for wild-type | 2053 [d] U/mg | [43] |
| *Horseradish* | mutant HRP 3-17E12 | X-33; pPICZαB | *AOX1* | 0.5% MeOH | ss of α-factor | Same as for wild-type | 1049 [d] U/mg | [43] |
| *Horseradish* | A2A isoenzyme | X-33; pPICZαC | *AOX1* | 0.5% MeOH | α-MF-pre-pro signal peptide | BMMY medium supplemented with 1% casamino acids and 1% sorbitol | 25.63 [a] U/mg | [65] |
| *Horseradish* | HRP-SpG | PpFWK3; pPpT4_alpha_S | *AOX1* | MeOH | NR | 136 h of methanol induction | 113 [d] mg/L | [66] |

**Table 2.** *Cont.*

| Source | Variant of Oxidoreductase Enzyme | Host Strain; Vector | Promoter | Inducer | Signal Sequence | Additional Information | Enzyme Yield | Ref. |
|---|---|---|---|---|---|---|---|---|
| **LAC** | | | | | | | | |
| *C. cinerea* | Lcc9 | X33; pGAPZαA | *GAP* | 0.5% glucose | ss of α-factor | Shake flasks 0.25 L; 0.3 mM CuSO₄; 20 °C; 4 days; 0.8% alanine | 12.8 [f] µkat/L | [67] |
| *P. ostreatus* | rPOXA 1B | X33; pGAPZαA | *GAP* | 0.5% glucose | ss of α-factor | Bioreactor 10 L; 1 mM CuSO₄; 2% peptone; 1.5% yeast extract; 170 h; geometry of flask | 3159.93 [f] U/L | [68] |
| *T. versicolor* | Lcc1 | SMD 1168; pHIL-D2 | *AOX1* | 0.5% MeOH | | Shake flasks 1 L; 0.1 mM CuSO₄; 20 °C; 3 days of induction | 11,500 [f] U/L | [69] |
| *T. versicolor* | Lcc1 | SMD 1168; pHIL-D2 | *AOX1* | 0.5% MeOH | | BioFlo III fermentor; 0.1 mM CuSO₄; 20 °C; 8.5 days | 140 [f] U/L | [69] |
| *T. versicolor* | Lcc1 | GS115; pPIC3.5 | *AOX1* | 1% MeOH | | Shake flasks (0.1 L) 0.2 mM CuSO₄; 22 °C; initial pH 6; 0.8% alanine: | 23.9 [f] U/L | [70] |
| *T. versicolor* | LccA | X33; pPICZαB | *AOX* | 0.6% MeOH | ss of α-factor | Shake flask (0.05 L of medium); 0.5 mM CuSO₄; 28 °C; 16 days; initial pH 7; | 11.972 [f] U/L | [71] |
| *T. versicolor* | LccA | X33; X33; pPICZαB | *AOX* | 0.6% MeOH | ss of α-factor | 5 L fermenter; 0.5 mM CuSO₄; 28 °C; 4.2 days; initial pH 7; | 18.123 [f] U/L | [71] |
| *C. gallica* | LcCg | X33; pPICZB | *AOX* | 1% MeOH | Modified α-factor preproleader | Fernbach flask; 0.5 mM CuSO₄; 28 °C; 12 days; initial pH 6; 0.8% alanine | 250 [e] U/L | [72] |
| *Trameters sp. 48424* | Lac48424-1 | GS115; pPIC3.5K | *AOX* | 0.5% MeOH | Native | Shake flasks; 0.3 mM CuSO₄; 20 °C; 7 days; initial pH 6; 0.8% alanine | 104.45 [f] U/L | [73] |
| *C. cinerea* | Lcc9 | GS115; pPIC9K | *AOX* | 0.5% MeOH | Native | Shake flasks 0.5 L; 0.3 mM CuSO₄; 28 °C; 10 days; initial pH 6.5; 0.8% alanine | 3138 ± 62 [f] U/L | [74] |
| *C. cinerea* | Lcc9 | X33; pPICZαA | *AOX* | 0.5% MeOH | ss of α-factor | Shake flasks 0.25 L; 0.3 mM CuSO₄; 20 °C; 7 days; 0.8% alanine | 9.3 [f] µkat/L | [67] |

NR, not reported; Ref., reference; ss, secretion signal; SpG, streptococcal protein G. [a] GOx/HRP activity was determined using o-dianisidine. [b] GOx activity was determined using 2,2′-azino-bis(3-ethylbenzothiazoline-6-sulphonic) acid (ABTS). [c] CDH activity was determined using 2,6-dichlorophenolindophenol (DCIP). [d] HRP activity was determined using 2,2′-azino-bis(3-ethylbenzothiazoline-6-sulphonic) acid (ABTS). [e] LAC activity was determined using syringaldazine (SGZ). [f] LAC activity was determined using 2,2′-azino-bis(3-ethylbenzothiazoline-6-sulphonic) acid (ABTS).

### 2.2.1. GOx

The yield of recombinant protein production in *P. pastoris* depends on several factors, such as gene optimization and synthesis, plasmid construction, host strain, and fermentation conditions [75].

As reported by Qui et al. (2016), the highest activity of recombinantly produced GOx was obtained when the expression was carried out at 30 °C and the pH of the medium was 6 [59,60].

Methanol is used as a source of energy and carbon for recombinant protein production in *P. pastoris* as an expression system. As a result, the methanol content in the culture medium is a significant variable while producing GOx [61,62].

Apart from the mentioned fermentation conditions, codon optimization can influence protein expression. It has been shown that replacing low-usage codons (<15%) with high-usage codons can increase the expression yield. For example, high-level expression of *P. notatum* F4 GOx was enabled via the distribution of G + C codons and removal of AT-rich regions, and the newly designed gene was named god-m [76].

The construction of multi-copy or high-copy transformants, which increase the gene dosage for expression GOx in *P. pastoris*, represents good strategies for enlarging the yield of production enzymes. Recent works have demonstrated that, to construct recombinant strains containing multiple copies of the GOx gene, it is first necessary to build vectors containing two or three tandem copies of the GOx expression cassette, termed pPICZHisα2GOX and pPICZHisα-3GOX. The generation of vectors containing more than three copies of the GOx expression cassette was unsuccessful, probably due to vector size

limitations. The best results are obtained for the secretory expression of GOx in a three-copy strain [77].

Research showed that the yield of expression of GOx can be regulated via the overexpression of the Hac1p, unfolded protein response regulator of chaperones. Yu et al. finally achieved an enzyme activity level of around 2125.3 U/mL in fermenter culture using basic process changes [77].

### 2.2.2. CDH

*P. pastoris* is established as a standard expression system for CDH. Bey et al. reported that cultivation mode impacts CDH expression; cultivation in a shake flask and a bioreactor produced 1176 U/L and 7800 U/L, respectively [63]. A higher yield of CDH is expected when multiple copy transformants are selected [63]. According to prior research, the duration of CDH expression depends on the CDH origin [39,64].

### 2.2.3. HRP

HRP expression in *P. pastoris* was achieved using the construct pPICZαB-HRP and, as an inducer, methanol. For better yields of that recombinant production, 1.0 mM vitamin B1, 1.0 mM δ-ALA, and 0.5 mL/l trace element mix (0.5 g/L MgCl2, 30 g/L $FeCl_2 \cdot 6H_2O$, 1 g/L $ZnCl_2 \cdot 4H_2O$, 0.2 g/L $CoCl_2 \cdot 6H_2O$, 1 g/L $Na_2MoO_4 \cdot 2H_2O$, 0.5 g/L $CaCl_2 \cdot 2H_2O$, 1 g/L $CuCl_2$, and 0.2 g/L $H_2BO_3$) were added to growth medium at the point of induction [43]. One such carbon source given to the medium is 1% sorbitol. Casamino acids are used to stop protein breakdown [65]. Specific productivity (qp) of the recombinant *P. pastoris* strain was increased 5.5-fold via a dynamic feeding strategy, where the setpoint for the specific substrate uptake rate (qs) was increased stepwise until a predetermined maximum (qsmax) was achieved, in contrast to a traditional feed-forward strategy [78,79]. *P. pastoris* successfully secreted a specific fusion construct of HRP and SpG in bioreactor cultivations with 110 mg/L yields. Recombinant HRP yields from *P. pastoris* that have been previously reported are normally in the 10 mg/L range [66].

### 2.2.4. LAC

Different strategies can be applied to achieve higher yields or higher activity of recombinantly produced LAC in *P. pastoris*. Concrete data is presented in Table 2. Previous studies have shown that there are two main aspects of optimizing LAC production in *P. pastoris* and yeast in general: control at the level of recombinant gene construction and control of the parameters to obtain optimal cultivation [80]. Prior research recognized temperature, methanol concentrations, pH value during cultivation, and copper concentration as important factors affecting heterologous LAC expression. Hong et al. concluded that cultivation in shake flasks at low temperatures with a final concentration of methanol 0.5% improved laccase activity; also, they confirmed their observation via cultivation under controlled conditions in the fermenter [69]. As has previously been reported in the literature, the expression of *T. versicolor* LAC, lcc1 in *P. pastoris*, was enhanced if the pH was periodically altered to pH 6.0 during fermentation [81]. O'Callaghan et al. designed a medium for maintaining a constant pH value. They examined the impact of proteolytic and non-proteolytic *P. pastoris* strains and the effect of copper availability on the laccase expression. They did not observe significant differences between proteolytic and non-proteolytic strains, but adding alanine to maintain pH and appropriate copper concentration had a significant impact [70]. Other researchers discovered that, via the optimization of several parameters such as the initial pH, the final concentration of methanol as an inducing agent, medium volume, initial OD600, copper concentration, and peptone concentration in the medium (pH 7, 0.6% MeOH, 50 mL of medium, 0.5 mM $Cu^{2+}$ and 4% peptone), researchers can obtain 4.4 times higher LAC activity compared to the initial medium. Besides that, they have also suggested that the native LAC secretion signal was superior to the secretion signal of the α-factor [71]. Difficulty detecting laccase associated with poor secretion using an α-factor secretion signal can be overcome using evolved α-factor preproleader [72]. According to Xu et al., the

significance of the laccase expression factors took the following order: concentration of methanol added > initial pH > $Cu^{2+}$ concentration > temperature [74]. A considerable body of literature promotes distinct optimal conditions for heterologous laccase expression. Ardila-Leal et al. statistically improved culture media for rPOXA 1B laccase production, expressed in *Pichia pastoris* containing pGAPZαA-LaccPost-Stop in a 10 L bioreactor [68]. They obtained 3159.93 U/L using ABTS as a substrate.

### 3. State-of-the-Art Technologies for Increasing Recombinant Protein Expression

In the articles mentioned above, the optimization of fermentation was achieved by changing the fermentation media composition, induction time, temperature optimization, etc. Still, there are also trials increasing the fermentation yield of these enzymes by using state-of-the-art technologies such as precision fermentation [82], directed evolution [83], protein and strain engineering [84], high-throughput screening methods based on in vitro compartmentalization [85], flow cytometry, and microfluidics.

### 3.1. Directed Evolution and Protein and Strain Engineering

Strain engineering for protein production can be performed using precision fermentation that combines synthetic biology, genetic engineering, and machine learning approaches. This is based on biofoundries that provide an integrated infrastructure for rapid construction, design, and analyzing genetically modified organisms [86]. The first step usually involves generating large host organism libraries via diverse genetic modifications like protease knock-out, cassette modifications, etc. Afterwards comes the screening process. Computational approaches such as deep learning based on artificial neural networks and analyzing genome sequences to predict gene manipulations to enhance recombinant protein production have recently been used [87–89]. This approach can predict the production performance of well-studied organisms like *S. cerevisiae* [90] and *P. pastoris* [91] and optimize metabolic flux via altering genes involved in the metabolic network [92]. Therefore, the machine learning approach is proven capable of recommending strain engineering strategies [92].

Adaptive laboratory evolution is another approach for improving microbial phenotypes in many organisms [93]. In this approach, microbes are cultured in a desired growth environment for an extended period, allowing natural selection to enrich mutant strains. The evolved strains are later characterized and their DNA is sequenced to find adaptive mutations that enable phenotypic improvement.

The expression of recombinant proteins in *S. cerevisiae*, especially oxidoreductases, can be increased using synthetic biology methods by choosing suitable promotors, selectable markers, and plasmids. Further, an increase in enzyme production can also be enhanced by utilizing various secretion factors. For example, it can be increased dramatically via site-directed mutagenesis or directed evolution of secretion peptide recombinant protein production. For instance, protein engineering approaches were carried out by Aza et al. to facilitate the heterologous production of various laccases by S. cerevisiae that included best-evolved signal peptides, new N-glycosylation sites in the enzyme genes, and consensus enzyme design to enhance protein folding and stability [94]. The introduction of N-glycosylation sites is case specific since it can lead to decreased activity but also can enhance protein folding and, therefore, the enzyme activity. Authors obtained mutated α-factor preproleader $\alpha_{9H2}$ that enhanced LAC production in the yeast twofold. Using other above-mentioned protein engineering strategies, they obtained 37 mg/L of ascomycete LAC. The same authors in another publication designed an improved universal signal peptide $\alpha_{OPT}$ by adding four mutations into the $\alpha_{9H2}$ preproleader sequence [52].

*P. pastoris* was used by Zhou et al. in 2023 for the expression of GOx; by screening different signal peptides, introducing multiple copies of genes, and engineering vesicle trafficking, the hyperproducing strain G1Ese (co-expressing trafficking components EES and SEC) was obtained that could produce up to 7223 U/mL with 30.7 g/L of GOx—that is 3.3 fold higher than the highest level reported so far [95]. It is also possible to engineer the

*P. pastoris* strain via co-expression of chaperons and protein disulfide isomerase in these yeast cells [96]. To increase the secretory expression of heterologous proteins in *P. pastoris*, Duan et al. screened endogenous signal peptides and protein folding factors. Their effects on the expression of three reporter proteins were tested and they were able to identify the Msb2 signal peptide and Dan4 signal peptide, both of which increase recombinant protein secretion 8 and 172 fold, respectively, compared to the alpha-mating preproleader sequence in *P. pastoris* [97].

Ito et al. l., in their recent work, created a terminator catalog by testing 72 sequences of terminators from *S. cerevisiae* and *P. pastoris* and found that terminator RNA sequences from *S. cerevisiae* maintain function when transferred to *P. pastoris* [98]. They managed to fine-tune protein expression levels in metabolic engineering and synthetic biology in *P. pastoris* and enhance them 17-fold. In a similar work on RNAi expression tuning, Wang et al. found genes with functions in cellular metabolism, protein modification, and degradation and a cell cycle that can significantly influence the expression level of proteins in *S. cerevisiae* [99].

One of the problems with expressing recombinant proteins in Pichia can be glycosylation, although it is a much bigger problem for *S. cerevisiae* and usually glycosylation is necessary for eukaryotic proteins to be correctly folded and expressed. To solve this problem, strain glyco-engineering trials of Pichia were performed to prevent hyperglycosylation and enable a higher fermentation yield of recombinant peroxidases [100]. Different glycoengineered *P. pastoris* strains were developed, and the physiology and growth behaviors of Man5GlcNAc2 glycosylating *P. pastoris* strain in the controlled environment of a bioreactor were characterized using flow cytometry during the expression of the HRP C1A isoform of the enzyme. The HRP C1A isoform expressed in the novel glycoengineered *Pichia* strain had similar kinetic characteristics to the one expressed in the wild-type *Pichia* strain. Still, the thermal stability of the recombinant HRP was decreased due to the reduced glycosylation. Furthermore, the recombinant enzyme formation rate in the novel strain increased from 0.77 U/gh to 1.05 U/gh during fermentation.

### 3.2. High-Throughput Screening Methods

#### 3.2.1. Flow Cytometry

To follow the influence of various factors on protein production in *P. pastoris*, it is essential to be able to follow the physiological state of recombinant yeast cells. Hyka et al. quantified factors affecting the physiological state of recombinant *P. pastoris* Mut+ (methanol utilization-positive) by using a combination of staining with different fluorescent dyes and analysis via flow cytometry [101]. The authors found that cell vitalities could range from 5% to 95% in high-cell-density cultures with strain-producing HRP, depending on the influence of various stresses such as recombinant protein expression, high cell density, and pH. This quantitative assessment of the individual cells' physiology using flow cytometry enables the implementation of innovative concepts in bioprocess development. This is especially important because the paradigm assumes a uniform cell population and does not differentiate between individual cells whose state can only be followed by single-cell analysis. The conclusion was that only part of the cell population contributes to the recombinant protein production, and the objective should be to maintain productive cells over a long period for as long as possible.

Flow cytometry can also be used for the following expression and correct folding of active proteins like cytochrome c peroxidase when the recombinant protein is fused with a green fluorescent protein (GFP) [102].

#### 3.2.2. Microfluidics

Since the screening phase and early process development based on microtiter plates and flasks still represents a bottleneck due to the high cost and time-consuming procedures, Totaro et al. developed a screening protocol for *P. pastoris* clone selection based on the multiplexed microfluidic device using 15 µL cultivation chambers that were able to operate in perfusion mode and monitor dissolved oxygen content in the culture in a non-invasive

way [103]. Using a microfluidic platform, the authors identified the best producer clone after 12 h from inoculation and confirmed the results via lab-scale fermentation.

Microfluidics combined with flow cytometry were also used for high-throughput droplet screening and genome sequencing analysis to improve the amylase-producing *A. oryzae* strain [104]. In this work, 450,000 droplets were screened within two weeks, and a high-producing strain with 6.6-fold increased production was found.

### 3.2.3. In Vitro Compartmentalization

*In vitro*, compartmentalization is often used in protein engineering and can be made in a polydisperse format for single-cell experiments, or it can be made in a monodisperse format via microfluidics [105]. Microspheres made of soft materials are also used in protein engineering as an alternative to liquid compartments [106]. Both of these compartmentalization methods can be used to not only improve enzyme activity and stability but also production yield during fermentation; usually, the best way to do so is to perform directed evolution experiments using strains for production.

To optimize recombinant protein production in yeasts (*P. pastoris*), droplet microfluidics can be used to encapsulate (compartmentalize) large genetic libraries of strains within biocompatible gel beads that are engineered to selectively retain any recombinant proteins of interest by binding it via His tag usually used for labeling and purification; afterward, staining of secreted protein using fluorescent dyes occurs [107]. This platform can be used broadly for various proteins, including oxidoreductases. As proof of principle, authors found a *P. pastoris* strain that 5.7-fold increased recombinant cutinase production after screening more than $10^6$ genotypes.

Compartmentalization within double emulsion can also be used to optimize recombinant protein production instead of beads. *In vitro*, compartmentalization within a double emulsion of water in oil was performed using microfluidics and fluorinated oil. The fluorescent immunosensor quench-body detected the secreted recombinant protein (fibroblast growth factor 9), and clones with high protein secretion were detected via fluorimetry [108]. This method also shortens the development period of industrial strains for recombinant protein production.

### 4. Conclusions

The optimization of culture conditions represents one of the most used techniques to overcome the problem of low yield since the composition of the medium plays a significant role in the production of recombinant proteins. Establishing optimal reaction conditions such as pH and temperature is one of the critical steps for a higher yield of recombinant expression.

It can be concluded that, despite the problems with the expression of OXR in yeasts like *S. cerevisiae* and *P. pastoris* due to the necessity of adding transition metals (copper and iron) and metabolic precursors of FAD and heme during fermentation, there are various approaches to increase the expression yield of these enzymes. Some of them are optimizing fermentation conditions, the codon usage, using strong promoters and terminators, and multi-copy expression vectors. *P. pastoris* usually gives higher expression yields of proteins with lesser glycosylation levels compared to *S. cerevisiae*, which usually gives smaller expression yields of recombinant proteins, very high glycosylation levels, and microheterogeneity of expressed proteins. Still, there are always exceptions depending on the specific recombinant protein and used yeast strain.

As we could see from the literature recently, there are also new possibilities to further increase in fermentation yield that explore cutting-edge technologies such as directed evolution, protein and strain engineering, high-throughput screening methods based on in vitro compartmentalization, flow cytometry, and microfluidics.

**Author Contributions:** Conceptualization, M.C.P., M.S. and R.P.; writing—original draft preparation, M.C.P. and M.S.; writing—review and editing, R.P.; visualization, M.C.P. and M.S.; funding acquisition, R.P. All authors have read and agreed to the published version of the manuscript.

**Funding:** This research was funded by the Ministry of Science, Technological Development and Innovation of the Republic of Serbia, grant No. 451-03-47/2023-01/200168 (University of Belgrade-Faculty of Chemistry).

**Conflicts of Interest:** The authors declare no conflicts of interest. The funders had no role in the design of the study; in the collection, analyses, or interpretation of data; in the writing of the manuscript; or in the decision to publish the results.

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
