# Peer review of "State of the Art Technologies for High Yield Heterologous Expression and Production of Oxidoreductase Enzymes: Glucose Oxidase, Cellobiose Dehydrogenase, Horseradish Peroxidase, and Laccases in Yeasts P. pastoris and S. cerevisiae"

_fermentation, doi:10.3390/fermentation10020093_

Round 1

Reviewer 1 Report

Comments and Suggestions for Authors

The review “State of the art technologies for high yield heterologous expression and production of oxidoreductase enzymes: glucose oxidase, cellobiose dehydrogenase, horseradish peroxidase, and laccases in yeasts P. pastoris and S. cerevisiae” is devoted to consideration of modern approaches to creation of yeast strains producing enzymes. The authors dwell not only on methods for constructing strains with desired properties, but also pay attention to the peculiarities of cultivating the resulting strains and new technologies that can be used to further increase the efficiency of heterologous enzyme synthesis. At the beginning of the review, the authors give a description of those enzymes, the features of heterologous synthesis of which are discussed in the review. All the enzymes considered can be used in medicine, biotechnology, and other industries. Therefore, the review may be of interest to specialists in these fields.

The review can be published after correcting errors, and answer to questions and comments.

Remarks and questions:

1.       Table 1 is not informative enough. The information given in the table does not allow us to determine the optimal parameters. To do this, it should be possible to compare the enzyme production when using the same expression cassette under different cultivation conditions, or vice versa, different expression cassettes and the same cultivation conditions. It may be worth thinking about the information and the format of their placement in the table.

2.       Why is there no comparison of the data from [37] and [50], where the same expression cassettes were used, but different concentrations of the inductor?

3.       The results of work [40] and [41] using the native secretion signal of CDH are very different. Why were yeasts cultivated for 5 days at [40] and only 16 hours at [41]? Have you observed the dynamics of accumulation of the target protein? Have you assessed the degree of glycosylation of the protein?

4.       About the results of work [42]: Is HRP secreted or remains in the ER? What is known in this case about the carbohydrate component of the enzyme?

5.       The use of a secretion signal in expression cassettes for LAC production ensures the secretion of the enzyme and the associated glycosylation of the enzyme. Does the structure of the carbohydrate component of the enzyme synthesized by yeast differ from the structure of the carbohydrate component of the native enzyme? Can this circumstance affect the activity of the enzyme?

6.       Yeast promoter names should be given in italics.

7.       Yeast proteins should be labeled as follows: Aox1p, Aox2p.

8.       Check the location of punctuation marks: in the text, the dots should be located after the links to the cited article.

9.       The statement about the influence of the parameters of the culture medium, fermentation conditions on the yield of recombinant proteins (lines 267-270) is applicable not only in the case of Pichia pastoris yeast, but also for Saccharomyces yeast. Therefore, it is probably not necessary to talk about this here as a feature of the expression system of the yeast Pichia pastoris.

10.   Table 2 contains a lot of information that is quite difficult to compare in order to determine the effect of specific parameters on the level of production of target proteins. It may be worth considering modifying the table structure.

11.   How stable is the maintenance of multiple expression cassette insertions in the yeast Pichia pastoris?

12.   What is «SpG» in table 2 and on line 314?

13.    Based on the results shown in Tables 1 and 2, it is difficult to draw an unambiguous conclusion about the advantages of a particular expression system. Column «Enzyme yield» shows only the units of activity, but the activity of the enzyme may depend on the cultivation parameters and other factors, so the values given cannot be directly correlated with the yield of a heterologous protein.

14.   Lines 380-381: ”…to facilitate the heterologous production of various laccases by S. cerevisiae that included 380 best-evolved signal peptides, new N-glycosylation sites in the enzyme genes…”

Does the introduction of new glycosylation sites not change the properties of the enzyme?

15.   Lines 408-409: “One of the main problems with expressing recombinant proteins in Pichia is hyperglycosylation…”

Hyperglycosylation of proteins is a much more serious problem for Saccharomyces cerevisiae yeast than for Pichia pastoris. In S. cerevisiae, glycoproteins can contain up to 200 mannose residues, in Pichia pastoris- 14-18. Therefore, it is not entirely correct to indicate this problem specifically for the P. pastoris.

16.   In the "Conclusion" section, I would like to see not only general words about the possibilities of modern technologies for producing yeast strains synthesizing heterologous enzymes, but also specific conclusions on comparing the expression systems of two types of yeast.

Author Response

Reviewer 1

Comments and Suggestions for Authors

The review “State of the art technologies for high yield heterologous expression and production of oxidoreductase enzymes: glucose oxidase, cellobiose dehydrogenase, horseradish peroxidase, and laccases in yeasts P. pastoris and S. cerevisiae” is devoted to consideration of modern approaches to creation of yeast strains producing enzymes. The authors dwell not only on methods for constructing strains with desired properties, but also pay attention to the peculiarities of cultivating the resulting strains and new technologies that can be used to further increase the efficiency of heterologous enzyme synthesis. At the beginning of the review, the authors give a description of those enzymes, the features of heterologous synthesis of which are discussed in the review. All the enzymes considered can be used in medicine, biotechnology, and other industries. Therefore, the review may be of interest to specialists in these fields.

The review can be published after correcting errors, and answer to questions and comments.

We thank Reviewer 1 for commenting that the article may interest specialists in fermentation fields and for the remarks that significantly improved our article.

Remarks and questions:

  1. Table 1 is not informative enough. The information given in the table does not allow us to determine the optimal parameters. To do this, it should be possible to compare the enzyme production when using the same expression cassette under different cultivation conditions, or vice versa, different expression cassettes and the same cultivation conditions. It may be worth thinking about the information and the format of their placement in the table.

We thank Reviewer 1 for the comment and agree with this remark. Therefore, as suggested, Table 1 was reformatted so that it is possible to compare the enzyme production when using the same expression cassette under different cultivation conditions.

  1. Why is there no comparison of the data from [37] and [50], where the same expression cassettes were used, but different concentrations of the inductor?

As Reviewer 1 suggested data from ref 50 (now 38). were included into the Table 1 and compared with the data from ref 37 (now 36).

  1. The results of work [40] and [41] using the native secretion signal of CDH are very different. Why were yeasts cultivated for 5 days at [40] and only 16 hours at [41]? Have you observed the dynamics of accumulation of the target protein? Have you assessed the degree of glycosylation of the protein?

The results of work [40] (now 39) and [41] (now 40) both using the native secretion signal of CDH are very different due to the use of different yeast strains. In work [40], the protease-free S. cerevisiae BJ5465 strain was used that can give high amounts of protein after long fermentation times. In contrast, in work [41], we used S. cerevisiae InvSc1 strain (Invitrogen), which usually gives a maximum of 1 mg/mL of recombinant protein after 16h of fermentation and is mostly used for directed evolution experiments.

Appropriate text was added (see lines 189-194).

  1. About the results of work [42]: Is HRP secreted or remains in the ER? What is known in this case about the carbohydrate component of the enzyme?

We added appropriate discussion about results of work [42] where HRP remains inside ER and after pre-propeptide sequence optimization export from ER was improved and deglycosylation of purified extracellular HRP was done to precisely determine amount of produced recombinant HRP (see lines 204-210).

  1. The use of a secretion signal in expression cassettes for LAC production ensures the secretion of the enzyme and the associated glycosylation of the enzyme. Does the structure of the carbohydrate component of the enzyme synthesized by yeast differ from the structure of the carbohydrate component of the native enzyme? Can this circumstance affect the activity of the enzyme?

We agree with Reviewer 1 that carbohydrate structure differs when expressed in yeasts and influences the stability and activity of the recombinant enzymes. Appropriate discussion was added to the main text (see lines 232-237).

  1. Yeast promoter names should be given in italics.

As Reviewer 1 suggested, yeast promoter names are given in italics. (text labeled in red).

  1. Yeast proteins should be labeled as follows: Aox1p, Aox2p.

As Reviewer 1 suggested, yeast proteins are labeled as follows: Aox1p, Aox2p (text labeled in red).

  1. Check the location of punctuation marks: in the text, the dots should be located after the links to the cited article.

As Reviewer 1 suggested, we changed the location of punctuation marks in the text so that they are located after the links to the cited article.

  1. The statement about the influence of the parameters of the culture medium, fermentation conditions on the yield of recombinant proteins (lines 267-270) is applicable not only in the case of Pichia pastoris yeast, but also for Saccharomyces yeast. Therefore, it is probably not necessary to talk about this here as a feature of the expression system of the yeast Pichia pastoris.

As suggested by Reviewer 1, The statement about the influence of the parameters of the culture medium, and fermentation conditions on the yield of recombinant proteins (lines 267-270) was transferred to the conclusion section since it can be applied for both yeasts (see Lines 490-494).

  1. Table 2 contains a lot of information that is quite difficult to compare in order to determine the effect of specific parameters on the level of production of target proteins. It may be worth considering modifying the table structure.

We thank Reviewer 1 for the comment and agree with this remark. Therefore, as suggested, Table 2 was reformatted so that it is possible to compare the enzyme production when using the same expression cassette under different cultivation conditions.

  1. How stable is the maintenance of multiple expression cassette insertions in the yeast Pichia pastoris?

The stability of genes integrated into the Pichia genome was discussed as suggested (see lines 265-266).

  1. What is «SpG» in table 2 and on line 314?

SpG is Streptococcal protein G and legend was updated in Table 2 (see line 271).

  1. Based on the results shown in Tables 1 and 2, it is difficult to draw an unambiguous conclusion about the advantages of a particular expression system. Column «Enzyme yield» shows only the units of activity, but the activity of the enzyme may depend on the cultivation parameters and other factors, so the values given cannot be directly correlated with the yield of a heterologous protein.

As was suggested by Reviewer 1, Tables 1 and 2 were reorganized according to the expression system to help conclude the advantages of a particular expression system. Enzyme yield shows units of activity since it is the only important parameter for the production of enzymes and their use as biocatalysts for substrate conversion to products (kcat).

  1. Lines 380-381: ”…to facilitate the heterologous production of various laccases by S. cerevisiae that included 380 best-evolved signal peptides, new N-glycosylation sites in the enzyme genes…”

Does the introduction of new glycosylation sites not change the properties of the enzyme?

Yes, it is case-specific since it can decrease activity due to the increase in molecular weight of the protein. Still, it can also increase activity due to helping protein folding. Appropriate discussion was added to the main text (see lines 397-399).

  1. Lines 408-409: “One of the main problems with expressing recombinant proteins in Pichia is hyperglycosylation…”

Hyperglycosylation of proteins is a much more serious problem for Saccharomyces cerevisiae yeast than for Pichia pastoris. In S. cerevisiae, glycoproteins can contain up to 200 mannose residues, in Pichia pastoris- 14-18. Therefore, it is not entirely correct to indicate this problem specifically for the P. pastoris.

Reviewer 1 is right. This problem is much more pronounced with S. cerevisiae, especially due to the microheterogeneity of glycosylation (50% per mass), and not so much for Pichia (30% per mass), so the statement was undermined as a problem in Pichia and mentioned as a bigger problem in S. cerevisiae (lines 424-425)

  1. In the "Conclusion" section, I would like to see not only general words about the possibilities of modern technologies for producing yeast strains synthesizing heterologous enzymes, but also specific conclusions on comparing the expression systems of two types of yeast.

As suggested by Reviewer 1 we added appropriate discussion in the “Conclusion section” on comparing the expression systems of the two types of yeast (lines 502-506).

Reviewer 2 Report

Comments and Suggestions for Authors

The paper of Popovic et al. is an interesting review dealing with the expression of  some oxidoreductase enzymes in yeasts. The work is well done and up to date and is suitable for publication. However there are some minor questions that should be solved:

line 141  "numerous inhibitors"  what are ? and what is their relevance in this context ?

line 144: GAP, TEF1 , GAL1  they are genes and should be written in italics ( GAP, TEF1, GAL1  ).

line 146  the phrase could be improved " i.e.  a low level of protein secretion, hyperglycosilation and...."

Table 1 : is not clear what is the column " Oxidoreductase", the name?, a mutant? a variant?
in line 3 ref 38  A. oryzae    2850 GAL-10   the inducer was galactose. Also in Table the name of genes shoud be in italic.

line 168  the indicated reference (50) is very relevant why was not inserted in Table 1 ?

line 173 S.cerevisiae should be in italic; GAL1 was not a galactose dehydrogenase, bat is a galactokinase and GAL 10 is an UDP-glucose-epimerase

Line 222  The authors should indicate here that now Pichia has been renamed as  Komagataella phaffii

Line 238-9  the phrase is wrong since the glycosilation in Pichia is  not similar to mammalian cells !!

In references: please correct the name of  all cited species:  The genus required the uppercase while the specie is in lowercase (es Neurospora crassa..), in addition all the name od species should be in italics ( Saccharomyces cerevisiaeTrametes versicolor..etc)

Ref 96 the page numbers are lacking  (3691-3707)

Ref 102 should be  " Proc Natl Acad Sci USA".

Author Response

REVIEWER 2

The paper of Popovic et al. is an interesting review dealing with the expression of  some oxidoreductase enzymes in yeasts. The work is well done and up to date and is suitable for publication. However there are some minor questions that should be solved:

We thank Reviewer 2 for commenting that the article is well done, up-to-date, and suitable for publication. We also thank you for the remarks that significantly improved our article.

  1. line 141  "numerous inhibitors"  what are ? and what is their relevance in this context ?

    As Reviewer 2 suggested, we added examples of the mentioned inhibitors that can decrease the fermentation yield of recombinant proteins (lines 141-143).
  2. line 144: GAP, TEF1 , GAL1  they are genes and should be written in italics ( GAP, TEF1, GAL1  ).

    As both reviewers suggested, the genes were written in italics (see changes labeled in red).
  3. line 146  the phrase could be improved " i.e.  a low level of protein secretion, hyperglycosilation and...."

    As Reviewer 2 suggested, the phrase was changed and improved accordingly (lines 147-148).
  4. Table 1 : is not clear what is the column " Oxidoreductase", the name?, a mutant? a variant?
    in line 3 ref 38  A. oryzae    2850 GAL-10   the inducer was galactose. Also in Table the name of genes shoud be in italic.

As Reviewer 2 suggested, we added an additional explanation that oxidoreductase means variants of oxidoreductase enzymes. Also, the names of genes are now given in italics.

  1. line 168  the indicated reference (50) is very relevant why was not inserted in Table 1 ?

    As Reviewer 2 suggested, we added reference (50) (now 38) to Table 1.
  2. line 173 S.cerevisiae should be in italic; GAL1 was not a galactose dehydrogenase, bat is a galactokinase and GAL 10 is an UDP-glucose-epimerase

We thank Reviewer 2 for these remarks and changed the data concerning GAL1 and GAL10  according to the suggestion (lines 173-174).

  1. Line 222  The authors should indicate here that now Pichia has been renamed as  Komagataella phaffii

    As Reviewer 2 suggested, we indicated that now Pichia has been renamed as Komagataella phaffii (lines 240).
  2. Line 238-9  the phrase is wrong since the glycosilation in Pichia is  not similar to mammalian cells !!

According to Reviewer 2 suggestion, we changed the statement for mammalian cells to eukaryotic cells (line 257).

  1. In references: please correct the name of  all cited species:  The genus required the uppercase while the specie is in lowercase (es Neurospora crassa..), in addition all the name od species should be in italics( Saccharomyces cerevisiae,  Trametes versicolor..etc)

    Ref 96 the page numbers are lacking  (3691-3707)

    Ref 102 should be  " Proc Natl Acad Sci USA".

As Reviewer 2 suggested, we corrected all cited species' names (see text labeled in red).

Round 2

Reviewer 1 Report

Comments and Suggestions for Authors

The authors answered the reviewer’s questions and supplemented the text of the manuscript, which allows us to better understand and evaluate the results obtained. The article may be accepted for publication